# Anti-Tumor Effects of Heat-Killed *L. reuteri* MG5346 and *L. casei* MG4584 against Human Colorectal Carcinoma through Caspase-9-Dependent Apoptosis in Xenograft Model

**DOI:** 10.3390/microorganisms10030533

**Published:** 2022-02-28

**Authors:** Suk-Jin Kim, Chang-Ho Kang, Gun-Hee Kim, Hyosun Cho

**Affiliations:** 1Department of Bio-Health Convergence Major, Duksung Women’s University, Seoul 01369, Korea; kimsukjin333@naver.com; 2Mediogen Co., Ltd., Jecheon-si 27159, Korea; changho-kang@naver.com; 3Department of Pharmacy, Duksung Women’s University, Seoul 01369, Korea; 4Duksung Innovative Drug Center, Duksung Women’s University, Seoul 01369, Korea

**Keywords:** colorectal cancer, parabiotics, xenograft, apoptosis

## Abstract

In this study, we examined the anti-tumor effects of heat-killed *Bifidobacterium* and *Lactobacillus* strains on human colorectal carcinoma RKO cells in in vitro and in vivo xenograft models. First, the cytotoxic and apoptotic effects of 11 different strains were examined using an MTT assay and flow cytometry, respectively. Then, xenograft BALB/c nude mice were implanted with RKO cells and orally administered with single or mixed heat-killed bacterial strains to examine their inhibitory effects on tumor growth. Additionally, the levels of cleaved caspase-9, -3, and -7 and PARP in tumor tissues were analyzed using Western blotting or immunohistochemistry staining. The results showed that RKO cells were highly susceptible to heat-killed *B. bifidum* MG731 and *L. reuteri* MG5346 and that *L. casei* MG4584 induced apoptosis to a greater extent than other strains. The oral administration of individual MG731, MG5346, or MG4584 significantly delayed tumor growth, and mixtures of MG5346 and MG4584 or MG731, MG5346, and MG4584 synergistically inhibited the tumor growth in the xenograft model. The expression of cleaved caspase-3, -7, and -9 and PARP in the tumor tissues was increased in Western blotting, and the expression of cleaved caspase-3 and PARP in immunohistochemistry staining was also increased. Therefore, we suggest that the use of the combination of MG5346 and MG4584 as parabiotics could effectively inhibit the growth of colorectal cancer.

## 1. Introduction

Parabiotics are non-viable microbial cells, crude cell extracts, or inactivated probiotics that confer benefits on human or animal consumers when administered in adequate amounts [1]. Heat-killed *Bifidobacterium* or *Lactobacillus* are representative parabiotic strains that reportedly exert anti-inflammatory, antibacterial, and immunomodulatory effects [2,3,4]. Parabiotics can be an alternative to probiotics for high-risk patients or those with underlying conditions that make it difficult to take live bacteria [5,6]. Compared to probiotics, parabiotics have characteristics such as a safe profile, long shelf life, stability within the digestive system, resistance to hydrolysis, and non-toxicity [7]. Recently, as interest in the application of parabiotics as pharmaceuticals has increased, research on the use of parabiotics for cancer prevention and treatment has been continuous [8]. Many in vitro studies have reported that heat-killed bacteria show significant cytotoxicity in cancer cells via caspase-mediated apoptosis [9]. Heat-killed *L. plantarum* I-UL4, *L. brevis*, and *L. paracasei* exhibit significant selective cytotoxicity in breast and colorectal cancer (CRC) cells by inhibiting proliferation and inducing apoptosis [10,11]. In particular, heat-killed *L. brevis* and *L. paracasei* inhibit the growth of human colorectal HT-29 cells through the induction of apoptosis [12]. Cell-free supernatants of *B. bifidum* are effective in combating cancer cells and are associated with a substantial improvement in gastrointestinal cancer [13]. In addition to surgery, radiotherapy, and chemotherapy, parabiotics can be one of the major alternatives for treating various cancers.

Colorectal cancer is the third most common cancer in the world, with 1.80 million reported cases and an estimated 881,000 deaths in 2018, according to the World Health Organization 2019 report [12]. Current chemotherapy drugs for CRC such as oxaliplatin and 5-fluorouracil (5-Fu) come with a variety of undesirable side effects, such as diarrhea, bone marrow suppression, peripheral neuropathy, and cardiotoxicity [14,15]. Parabiotics distinguish between normal cells and cancer cells by modulating the proliferation of normal cells but driving apoptosis in cancerous cells.

The induction of apoptosis in cancer cells is mediated by the activation of several signaling molecules and caspases. Apoptosis can be initiated either through the extrinsic or intrinsic pathway. The extrinsic pathway begins with the binding of TNF-α to TNFR1 or interaction between Fas and FasL, which leads to the activation of caspase-8 through membrane-associated proteins TNF receptor-associated death domain (TRADD) and Fas-associated death domain protein (FADD) [16]. The intrinsic pathway is triggered by endogenous stimuli, such as DNA damage and oxidative stress. It is initiated by the release of cytochrome *c* from the mitochondrial intermembrane space into the cytoplasm, which leads to the cleavage and activation of a caspase cascade, specifically caspase-9, -3, and -7. Poly-ADP-ribose polymerase (PARP) is cleaved by caspase-3 and -7 and induces apoptosis [17].

This present study aimed to investigate the therapeutic possibilities of heat-killed *Lactobacillus* strains against human colorectal carcinoma. We initially used 11 different heat-killed *Bifidobacterium* and *Lactobacillus* strains to screen for cytotoxicity and apoptotic activity against human colorectal carcinoma RKO cells. In addition, we investigated the anti-tumor effects of selected heat-killed *Bifidobacterium* and *Lactobacillus* strains in xenograft animal models implanted with RKO cells. Moreover, we attempted to elucidate the molecular mechanism of the anti-tumor activity of heat-killed *Bifidobacterium* and *Lactobacillus* strains.

## 2. Materials and Methods

### 2.1. Bacterial Strain Preparation

Heat-killed bacterial strains were provided as powder by Mediogen Co., Ltd. (Jecheon, Korea). Cell pellets were obtained from bacterial cultures by centrifugation at 5000× *g* for 5 min and washed three times with distilled water. The bacteria were heat-killed at 100 °C for 30 min and then freeze-dried and stored at −20 °C until they were used. The following 11 strains were used: *B. bifidum* MG731, *B. breve* MG729, *L. bulgaricus* MG515, *L. casei* MG311, *L. casei* MG4584, *L. gasseri* MG4514, *L. plantarum* MG4215, *L. reuteri* MG5346, *L. rhamnosus* MG316, *L. rhamnosus* MG5200, and *Streptococcus thermophilus* MG5140.

### 2.2. Cell Culture

Human colorectal carcinoma RKO cells were obtained from American Type Culture Collection (ATCC; Manassas, VA, USA) and incubated in Dulbecco’s Modified Eagle’s medium (DMEM) supplemented with 10% fetal bovine serum (FBS) and 1% penicillin/streptomycin at 37 °C in a humidified atmosphere of 5% CO_2_. DMEM, FBS, and penicillin/streptomycin were purchased from Gibco (Grand Island, NY, USA).

### 2.3. Cell Cytotoxicity

The cytotoxicity of heat-killed bacterial strains against RKO cells was determined by measuring the cell viability using an MTT assay. RKO cells were plated in 96-well plates at a density of 5 × 10^3^ cells/well and incubated for 24 h. After the incubation, the cells were treated with the powder of heat-killed bacterial strains (1 × 10^8^ or 1 × 10^9^ cells/mL) for 24 h. Then, cells were added to MTT solution (1 mg/mL) and incubated for 4 h. The supernatant was then removed and 100 μL of dimethyl sulfoxide (DMSO; Tokyo Chemical Industry Co., Ltd., Tokyo, Japan) was added to each well to dissolve the formazan crystals. Absorbance was measured using a microplate reader (BMG Labtech, Offenburg, Germany) at 450 nm.

### 2.4. Cell Morphology

RKO cells were seeded at a density of 1 × 10^6^ cells/well in 6-well plates. After 24 h of incubation, cells were treated with the powder of heat-killed bacterial strains (1 × 10^9^ cells/mL) for 24 h. Cell morphology was then observed under a light microscope (200×; Nikon Eclipse TS100, Nikon, Tokyo, Japan).

### 2.5. Annexin V-FITC and Propidium Iodide Staining Assay

Cell apoptosis was detected using the FITC Annexin V Apoptosis Detection Kit I (BD Biosciences, San Diego, CA, USA). Briefly, RKO cells were cultured (1 × 10^6^ cells/well) overnight in 6-well plates. The cells were then treated with the powder of heat-killed bacterial strains (1 × 10^9^ cells/mL) for 24 h, collected, and added to Annexin V–FITC and propidium iodide (PI). After incubation for 15 min at room temperature in the dark, cells were immediately analyzed by a flow cytometer (NovoCyte Flow Cytometer, ACEA Biosciences Inc., San Diego, CA, USA). Four different cell populations were easily distinguished: unlabeled cells (live cells), cells stained with Annexin V–FITC only (early apoptotic cells), cells stained with PI (dead cells), and cells stained with both Annexin V—FITC and PI (late apoptotic cells).

### 2.6. Human Colorectal Cancer Xenografts in BALB/c Nude Mice

Animal experiments were conducted in accordance with the National Research Council’s Guide (IACUC, Seoul, Korea) for the Care and Use of Laboratory Animals. The experimental protocol was approved by the Animal Experiments Committee of Duksung Women’s University (permit number: 2021-005-005, approval date: 1 June 2021). BALB/c nude mice (female, five weeks old; Raonbio Co. Ltd., Seoul, Korea) were used as xenograft animal models. Mice were individually accommodated in a pathogen-free controlled environment (23–27 °C and 45 ± 5% humidity under a 12 h day/12 h night cycle). Mice were fed at random with standard laboratory chow and were provided with water ad libitum. RKO cells (1 × 10^6^ cells/mouse) were subcutaneously injected into the back, next to the right hind leg. The powder of heat-killed bacterial strains was dissolved in drinking water and administered orally every day for 3 weeks. Mice were randomly assigned to the following six groups (*n* = 8/group): (1) control group: drinking water; (2) MG731 group: *B. bifidum* MG731 (1 × 10^9^ cells/mouse, this choice of the concentration was based on in vitro studies); (3) MG5346 group: *L. reuteri* MG5346 (1 × 10^9^ cells/mouse); (4) MG4584 group: *L. casei* MG4584 (1 × 10^9^ cells/mouse); (5) 2Mix group: mixture of *L. reuteri* MG5346 (1 × 10^9^ cells/mouse) and *L. casei* MG4584 (1 × 10^9^ cells/mouse); (6) 3Mix group: mixture of *B. bifidum* MG731 (1 × 10^9^ cells/mouse), *L. reuteri* MG5346 (1 × 10^9^ cells/mouse), and *L. casei* MG4584 (1 × 10^9^ cells/mouse). Tumors were identified and measured every other day using a standard caliper; tumor volume was calculated as [tumor length (mm) × tumor width (mm)^2^]/2, as previously described [18,19]. After the tumor volume had reached 2000 mm^3^, animals were euthanized and tumors were harvested.

### 2.7. Western Blot Analysis

The levels of apoptosis-related proteins in tumor tissue were determined by Western blot analysis [18]. The specimens were lysed with an extraction buffer (Intron Biotechnology, Seoul, Korea) to extract proteins, and protein concentration was determined by Coomassie (Bradford) Protein Assay (genDEPOT, Katy, TX, USA). The proteins were separated by electrophoresis and transferred onto nitrocellulose membranes (0.45-µM pore size, Merck Millipore, Burlington, MA, USA). The membranes were blotted with the following primary and secondary antibodies: Apoptosis Sampler Kit (#9915), anti-caspase-8 Ab (#9746), anti-GAPDH (#5174), anti-rabbit IgG (#7074), and anti-mouse IgG (#7076), all obtained from Cell Signaling Technology (CST, Danvers, MA, USA). Proteins were visualized using chemiluminescence detection (FluorChem E system, ProteinSimple, San Jose, CA, USA) and quantified using the ImageJ program (National Institutes of Health, Bethesda, MD, USA). The expression levels relative to GAPDH were determined.

### 2.8. Immunohistochemistry (IHC)

Sections were stained following standard methods [18]. Tumor tissues excised from mice were frozen with Frozen Section Compound (Leica, Hesse, Germany) and stored at −80 °C. The tumor tissue blocks were cut into 4 μm-thick sections. The sections were then incubated overnight at 4 °C with anti-cleaved caspase-3 (#9664, CST) and anti-cleaved PARP antibodies (#5625, CST) at dilutions of 1:500 and 1:50, respectively, in TBST. The slices were incubated with the secondary antibody, 1:100 mouse anti-rabbit IgG (#7074, CST), at room temperature for 1 h. The slices were then stained with DAB (Vector Laboratories, Burlingame, CA, USA) and visualized under a microscope (200× or 400×).

### 2.9. Statistical Analysis

All data were obtained in triplicate and presented as mean ± standard deviation (SD). Statistical analyses were performed using one-way analysis of variance (ANOVA) with Duncan’s multiple comparisons. Data were analyzed with SPSS v22 (IBM Corp., Armonk, NY, USA). The values of *p* < 0.05 were regarded statistically significant.

## 3. Results

### 3.1. Cytotoxic Effect of Heat-Killed Bifidobacterium and Lactobacillus Bacterial Strains on Human Colorectal Carcinoma RKO Cells

The viability of RKO cells was significantly decreased by 1 × 10^9^ cells/mL of heat-killed *B. bifidum* MG731 (to 37.66%) and *L. reuteri* MG5346 (to 54.39%), which was supported by microscopic observations (Figure 1). The same concentrations of *L. bulgaricus* MG515, *L. casei* MG311, *L. casei* MG4584, and *L. gasseri* MG4514 showed moderate cytotoxic effects on RKO cells (Figure 1b).

### 3.2. Apoptotic Effect of Heat-Killed Bifidobacterium and Lactobacillus Strains on RKO Cells

Total apoptosis (%), defined as the sum of early apoptosis (%) and late apoptosis (%) (the Annexin V+/PI− and Annexin V+/PI+ quadrants, respectively, in Figure 2a), was dramatically increased by 1 × 10^9^ cells/mL of *L. rhamnosus* MG5200 (to 15.48%), *L. gasseri* MG4514 (to 14.69%), and *L. casei* MG4584 (to 12.50%) (Figure 2b). The other strains, except for *L. casei* MG311, also induced apoptosis to some degree.

### 3.3. Anti-Tumor Effects of Heat-Killed Bifidobacterium and Lactobacillus Strains in Xenograft Model Bearing RKO Cells

To confirm the anti-tumor effect of heat-killed *Bifidobacterium* and *Lactobacillus*, we orally administered 1 × 10^9^ cells/mouse of *B. bifidum* MG731, *L. reuteri* MG5346, *L. casei* MG4584, respectively, in single-strain groups of xenograft models. The 2Mix group was administered a mixture of *L. reuteri* MG5346 (1 × 10^9^ cells/mouse) and *L. casei* MG4584 (1 × 10^9^ cells/mouse), and the 3Mix group was administered a mixture of *B. bifidum* MG731 (1 × 10^9^ cells/mouse), *L. reuteri* MG5346 (1 × 10^9^ cells/mouse), and *L. casei* MG4584 (1 × 10^9^ cells/mouse). All the selected heat-killed *Bifidobacterium* and *Lactobacillus* strains and their mixtures significantly inhibited tumor growth in comparison with drinking water at day 19 (Figure 3a–c). In particular, the tumor volume was 1018.45 ± 52.59 mm^3^ in the MG731 group, 1095.08 ± 83.58 mm^3^ in the MG5346 group, 1050.56 ± 46.22 mm^3^ in the MG4584 group, 493.53 ± 52.19 mm^3^ in the 2Mix group, and 447.35 ± 104.96 mm^3^ in the 3Mix group (Figure 3b). Interestingly, the tumor volume in the 2Mix and 3Mix groups was dramatically reduced (by 41% and 35%, respectively) in comparison with the control group (*p* < 0.05). The MG4584 group showed a significant tumor growth inhibitory effect in comparison with the control group from day 9, earlier than the other single-strain treatment groups. The tumor weights in all groups (control group: 1.76 ± 0.39 g; MG731 group: 0.78 ± 0.29 g; MG5346 group: 0.78 ± 0.19 g; MG4584 group: 0.88 ± 0.33 g; 2Mix group: 0.38 ± 0.33 g; 3Mix group: 0.32 ± 0.22 g) were positively correlated with tumor volumes (Figure 3c). A steady increase in whole-body weight in all groups was recorded throughout the experimental period (Figure 3d), indicating that there was no general toxicity caused by the oral administration of *Bifidobacterium* or *Lactobacillus* heat-killed strains.

### 3.4. Caspase-9-Dependent Apoptosis in RKO Cell–Derived Tumors Induced by Heat-Killed Bifidobacterium and Lactobacillus Strains

We investigated the expression of signaling molecules related to apoptosis in the tumor tissues derived from RKO cells. The caspase cascades, including caspase-8, -9, -3, and -7 and PARP, are known to be associated with apoptosis in tumors [20]. The ratios of cleaved caspase-9 to caspase-9 in tumor tissues derived from all treatment groups were significantly higher than those in the control group, whereas those of caspase-8 differed little among all groups (Figure 4a–c). The ratios of cleaved caspase-3 and caspase-7 were also significantly higher in tumor tissues from all treatment groups than in those from the control group (Figure 4a,d,e). The ratios of cleaved PARP to PARP were also higher in tumor tissues from the MG5346, MG4584, 2Mix, and 3Mix groups than in those from the control group (Figure 4a,f).

We also confirmed the increase in the levels of cleaved caspase-3 and cleaved PARP by the IHC staining of tumor tissues derived from RKO cell–implanted xenograft mice that were administered 2Mix or 3Mix (Figure 5).

## 4. Discussion

The present study demonstrates the anti-tumor effect of heat-killed *Bifidobacterium* and *Lactobacillus* bacterial strains on human colorectal carcinoma RKO cells in in vitro and in vivo xenograft models. The anti-tumor effect of live *Bifidobacterium* or *Lactobacillus* strains has been reported in many studies, but the anti-tumor effects of the heat-killed strains are yet unclear, especially in human CRC cells. *B. bifidum* MG731, which had a strong anti-tumor effect in CRC when administered in combination with PD-1 [21], was used as a positive control in this study.

Previous studies have reported that the administration 2 × 10^9^ CFU of live probiotic strains in CRC patients for 12 weeks improved bowel symptoms and quality of life [22]. Moreover, the pre-surgical use of *L. acidophilus*, *L. casei*, *L. lactis*, *B. bifidum*, *B. longum*, and *B. infantis* mixture promotes the faster return of normal gut function and shorter duration of hospital stay when used in elective surgery in colorectal cancer patients [23]. Therefore, we used 1 × 10^8^ or 1 × 10^9^ cells to evaluate the cytotoxic effect of heat-killed strains on RKO cells. In Figure 1, we found a higher cytotoxic effect at 1 × 10^9^ cells/mL in most of the tested heat-killed strains, although there was some variability. Several studies have reported that *B. bifidum* 3440 extract has the highest cytotoxicity (40%) in comparison with *B. longum* 3128, *B. lactis* 5854, and *B. infantis* in human non-small cell lung cancer A549 cells [24]. The treatment of human gastric adenocarcinoma with probiotic *L. reuteri* showed cytotoxic effects of 74.4%, 66.7%, and 40.8% at the ratios of 1:10, 1:100, and 1:1000, respectively [25]. Similarly, in our study, the cytotoxicity of *B. bifidum* MG731 (37.66%) and *L. reuteri* MG5346 (54.39%) to human CRC cells was higher than that of other bacterial strains (Figure 1b). Apoptosis is a controlled event, usually accompanied by minimal loss of membrane integrity until the late stage or secondary necrosis [26]. We found that 1 × 10^9^ cells/mL of heat-killed *L. rhamnosus* MG5200, *L. gasseri* MG4514, and *L. casei* MG4584 significantly induced apoptosis (in about 15% of the cells; Figure 2b).

There was no precise correlation in numerical values between cytotoxicity and the apoptotic effect of any of the tested heat-killed bacterial strains in our study (Figure 1 and Figure 2). We speculate that cancer cell death can be triggered not only by apoptosis but also by other signaling pathways [27,28]. Additionally, the different ways of measuring cytotoxic activity and apoptosis could contribute to the numerical variability in our results. However, *B. bifidum* MG731 and *L. reuteri* MG5346, which showed highest cytotoxic activity, statistically significantly induced apoptosis, and *L. rhamnosus* MG5200, *L. gasseri* MG4514, and *L. casei* MG4584, which were ranked as the top three strains for apoptosis induction, showed strong cytotoxicity against RKO cells (Figure 1b and Figure 2b).

Our selection of heat-killed *B. bifidum* MG731, *L. reuteri* MG5346, and *L. casei* MG4584 for an in vivo xenograft model experiment was based on the above in vitro results as well as companies’ capability of the mass production of bacterial strains. In BALB/c nude mice subcutaneously injected with RKO, we found a significant inhibition in the growth of CRC xenograft tumors when selected single heat-killed bacterial strains or their mixtures were orally administrated (Figure 3). A single administration of each of the three strains inhibited tumor growth by 15.18%, and each mixture significantly suppressed tumor growth by more than 61.76% in comparison with the control group. Recently, live or heat-killed *L. casei* was reported to protect mice against DMH-induced CRC, which partly agrees with our in vivo study results [29,30]. For the first time, we report a synergistic inhibitory effect of heat-killed *L. casei* MG4584 and *L. reuteri* MG5346 on CRC growth in a xenograft model, which suggests that various combinations of parabiotics or probiotics could be screened for use as therapeutic agents against cancer.

Subsequently, we examined the anti-tumor mechanisms of selected heat-killed bacterial strains in our xenograft model. We found that the significant delay in tumor growth in xenograft mice treated with heat-killed *B. bifidum* MG731, *L. casei* MG4584, and *L. reuteri* MG5346 was mediated by the increased levels of caspase-9, -3, and -7 and PARP in tumor tissues derived from xenograft (Figure 4). Caspases, cysteine aspartate proteases, are major regulators of apoptosis and are classified into initiator caspases and executive caspases [31]. The initiator caspases, caspase-8 and -9, are activated by death-inducing signals and subsequent signals that induce the cleavage of caspase-3, -7, and PARP, which are the executive and terminator caspases [32]. The cleavage of caspases induces apoptosis and leads to cell death. A recent study reported that the injection of a *Lactobacillus*-derived siderophore inhibited 40% of tumor growth in a xenograft model through apoptosis, which was mediated by the cleavage of caspase-3 and PARP [28], which agrees with our data in Figure 5. Many in vitro studies suggest that *B. bifidum* extracts increase the levels of cleaved caspase-3 and cleaved PARP in A549 and H1299 cells, consequently inducing apoptosis [21]. *L. rhamnosus* induces apoptosis through the mitochondrial pathway, as indicated by cytochrome *c* release and the cleavage of caspase-9 and -3 in human colorectal Caco-2 cells [33]. Similarly, we found that heat-killed *L. casei* MG4584 and *L. reuteri* MG5346 induce intrinsic apoptosis through the activation of caspase-9, which successively activates caspase-3 and -7 and PARP. Interestingly, heat-killed *L. casei* MG4584 and *L. reuteri* MG5346 did not induce the activation of caspase-8, which was also reported by Altoncy et al. [33]. Heat-killed *Lactobacillus* have cell wall components such as (lipo)teichoic acids, neutral and acidic polysaccharides, peptidoglycan, and (surface) proteins. Another recent study reported that peptidoglycan of *L. paracase* has anti-cancer activity by inducing caspase-3 activation in vitro and that the combination of *Bifidobacteria* lipoteichoic acid and 5-FU significantly inhibits tumor proliferation and induces apparent apoptosis [34,35]. In this study, it is assumed that the anti-tumor effect of the heat-killed *L. casei* MG4584 and *L. reuteri* MG5346 is due to lipoteichoic acid, peptidoglycan, and polysaccharides, etc., and further studies are needed to confirm this.

## 5. Conclusions

Our study demonstrates that heat-killed *Bifidobacterium* and *Lactobacillus* strains induce the intrinsic apoptosis of human colorectal carcinoma RKO cells in vitro and that heat-killed *L. casei* MG4584 and *L. reuteri* MG5346 have strong anti-tumor effects in an RKO cell-derived xenograft model through the activation of caspase-9, -3, and -7 and PARP. Our findings make a strong case for the use of heat-killed *L. casei* MG4584 and *L. reuteri* MG5346 as anti-tumor agents in colorectal cancer.

## Figures and Tables

**Figure 1 microorganisms-10-00533-f001:**
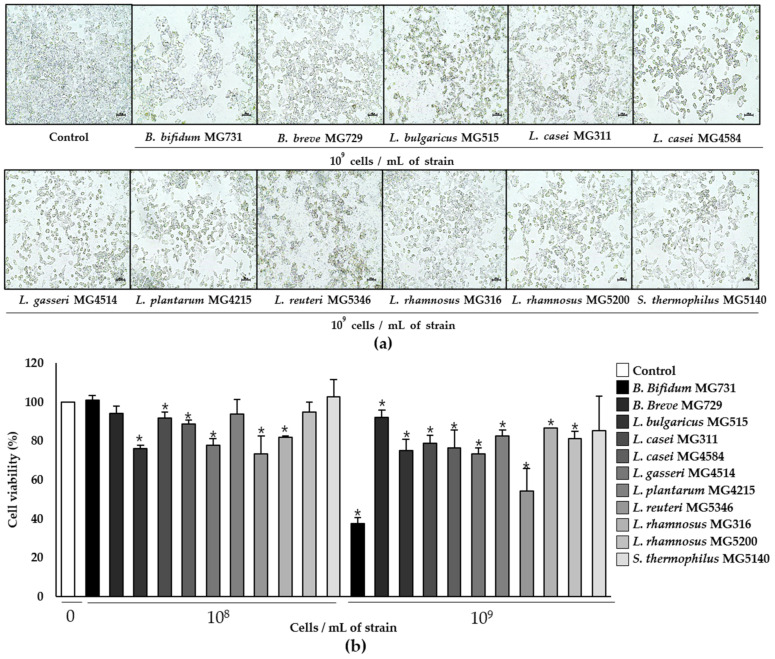
Cytotoxicity of heat-killed *Bifidobacterium* and *Lactobacillus* against human colorectal carcinoma RKO cells analyzed by MTT assay. (**a**) Cell morphology and (**b**) cell viability (%). The scale bar represents 200 px. Data represent mean ± SD (*n* = 3), * *p* < 0.05 vs. control.

**Figure 2 microorganisms-10-00533-f002:**
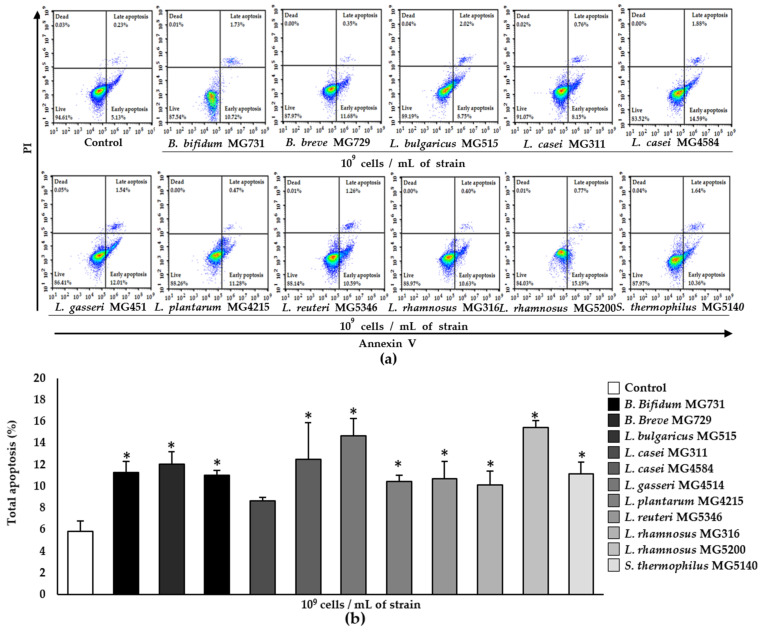
Apoptosis induced by heat-killed *Bifidobacterium* and *Lactobacillus* in RKO cells analyzed by flow cytometry. (**a**) Representative Annexin V and PI staining for apoptosis and (**b**) total apoptosis (%). Data represent mean ± SD (*n* = 3), * *p* < 0.05 vs. control.

**Figure 3 microorganisms-10-00533-f003:**
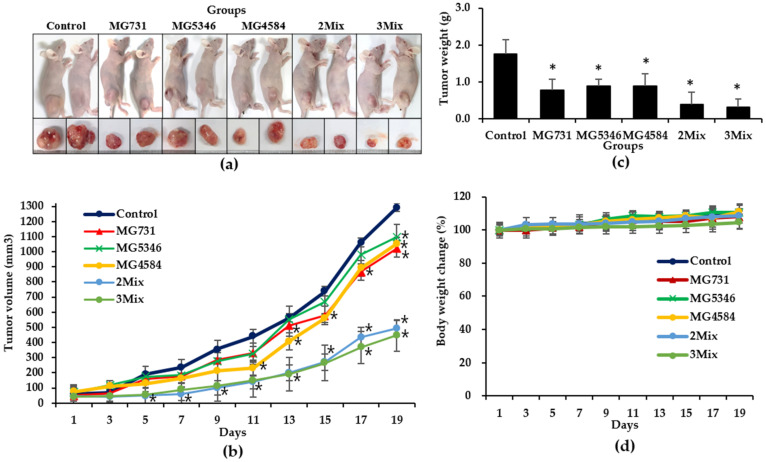
The inhibitory effect of heat-killed *Bifidobacterium* and *Lactobacillus* on the growth of RKO cell-derived tumors in xenograft model mice. BALB/c nude mice were subcutaneously injected with 1 × 10^6^ RKO cells/mouse into the dorsum next to the right hind leg. The mice were orally administered drinking water, heat-killed single strains, or their mixtures every day for another 19 days. (**a**) Photographs represent tumor size. (**b**) Tumor growth curves. (**c**) Tumor weight. (**d**) Body weight changes. Data represent mean ± SD (*n* = 5), * *p* < 0.05 vs. control. Control, drinking water; MG731, *B. bifidum* MG731 (1 × 10^9^ cells/mouse); MG5346, *L. reuteri* MG5346 (1 × 10^9^ cells/mouse); MG4584, *L. casei* MG4584 (1 × 10^9^ cells/mouse); 2Mix, mixture of *L. reuteri* MG5346 (1 × 10^9^ cells/mouse) and *L. casei* MG4584 (1 × 10^9^ cells/mouse); 3Mix, mixture of *B. bifidum* MG731 (1 × 10^9^ cells/mouse), *L. reuteri* MG5346 (1 × 10^9^ cells/mouse), and *L. casei* MG4584 (1 × 10^9^ cells/mouse).

**Figure 4 microorganisms-10-00533-f004:**
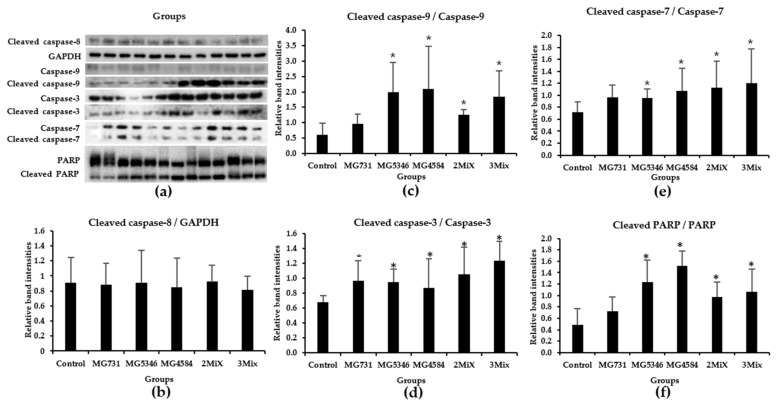
Protein levels in tumor tissues. After 19 days of treatment with heat-killed single strains or their mixtures, proteins were extracted from tumor tissues and analyzed by Western blotting assay. (**a**) Western blot image. (**a**–**f**) Western blot quantification: (**b**) cleaved caspase-9/caspase-9, (**c**) cleaved caspase-8/GAPDH, (**d**) cleaved caspase-3/caspase-3, (**e**) cleaved caspase-7/caspase-7, (**f**) cleaved PARP/PARP. Data represent mean ± SD (*n* = 6), * *p* < 0.05 vs. control. Control, drinking water; MG731, *B. bifidum* MG731 (1 × 10^9^ cells/mouse); MG5346, *L. reuteri* MG5346 (1 × 10^9^ cells/mouse); MG4584, *L. casei* MG4584 (1 × 10^9^ cells/mouse); 2Mix, mixture of *L. reuteri* MG5346 (1 × 10^9^ cells/mouse) and *L. casei* MG4584 (1 × 10^9^ cells/mouse); 3Mix, mixture of *B. bifidum* MG731 (1 × 10^9^ cells/mouse), *L. reuteri* MG5346 (1 × 10^9^ cells/mouse), and *L. casei* MG4584 (1 × 10^9^ cells/mouse).

**Figure 5 microorganisms-10-00533-f005:**
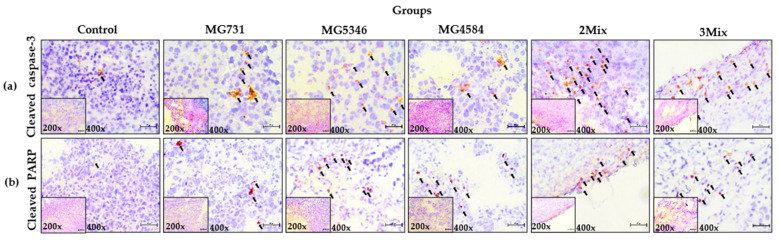
Cleaved caspase-3 and PARP in tumor tissues analyzed by immunohistochemistry (IHC). After 19 days of treatment with heat-killed single strains or their mixtures, tumor tissues were harvested and stained by IHC. (**a**) Cleaved caspase-3, (**b**) cleaved PARP. Black arrows indicate the dots of cleaved caspase-3 and cleaved PARP (magnification 20× or 40× as indicated). The scale bar represents 200 px. Control, drinking water; MG731, *B. bifidum* MG731 (1 × 10^9^ cells/mouse); MG5346, *L. reuteri* MG5346 (1 × 10^9^ cells/mouse); MG4584, *L. casei* MG4584 (1 × 10^9^ cells/mouse); 2Mix, mixture of *L. reuteri* MG5346 (1 × 10^9^ cells/mouse) and *L. casei* MG4584 (1 × 10^9^ cells/mouse); 3Mix, mixture of *B. bifidum* MG731 (1 × 10^9^ cells/mouse), *L. reuteri* MG5346 (1 × 10^9^ cells/mouse), and *L. casei* MG4584 (1 × 10^9^ cells/mouse).

## Data Availability

MDPI Research Data Policies.

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
