# Peer review of "Anti-Tumor Effects of Heat-Killed L. reuteri MG5346 and L. casei MG4584 against Human Colorectal Carcinoma through Caspase-9-Dependent Apoptosis in Xenograft Model"

_microorganisms, 2022, doi:10.3390/microorganisms10030533_

Round 1

Reviewer 1 Report

Kim SukJin et al. described in this manuscript addresses the anti-tumor effect of heat-killed Bifidobacterium and Lactobacillus using RKO cell line and xenograft Balb/c nude mice. The main manuscript structure is well described and will likely become a cited example of how to undertake such tasks.

In this manuscript, some modifications would be needed;

It would be good to have a discussion on why they administered the dead bacteria and did not consider administering the live bacteria preparation. If the anti-tumor effect is as good as that of the viable formulation, then perhaps the heat-sterilized formulation has considerable advantages in terms of storage, administration, and expiration date, and it would be good to emphasize and discuss this point more.

The image in Figure 5 is unclear, so a higher resolution image with more contrast is desirable.

Minor

P.9 L300

Heat-treated L. 323 casei MG4584 and L. reuteri MG5346 have been shown to have strong antitumor effects, but does authors think what in the extracts from these bacteria supports caspace and PARP activation?

Author Response

Reviewer #1

Kim SukJin et al. described in this manuscript addresses the anti-tumor effect of heat-killed Bifidobacterium and Lactobacillus using RKO cell line and xenograft Balb/c nude mice. The main manuscript structure is well described and will likely become a cited example of how to undertake such tasks.

In this manuscript, some modifications would be needed;

It would be good to have a discussion on why they administered the dead bacteria and did not consider administering the live bacteria preparation. If the anti-tumor effect is as good as that of the viable formulation, then perhaps the heat-sterilized formulation has considerable advantages in terms of storage, administration, and expiration date, and it would be good to emphasize and discuss this point more.

  • Thank you for the comments, we emphasized and added to the benefits of parabiotics.

Line 38-42: Parabiotics are an alternative to probiotics for high-risk patients or those with under-lying conditions that make it difficult to take live bacteria. Compared to probiotics, parabiotics have characteristics such as a safe profile, long shelf life, stability to diges-tive system, resistance to hydrolysis, and non-toxicity [5, 6].

The image in Figure 5 is unclear, so a higher resolution image with more contrast is desirable.

à As you suggested, we replaced the image in Figure 5 with a high-resolution image.

Minor

P.9 L300

Heat-treated L. 323 casei MG4584 and L. reuteri MG5346 have been shown to have strong antitumor effects, but does authors think what in the extracts from these bacteria supports caspace and PARP activation?

  • We assume that the anti-tumor effect of parabiotics is due to the constituents such as (lipo)teichoic acids, peptidoglycan, and polysaccharides etc., which is now descrided as followed.

Line 317-321: Heat killed Lactobacillus have cell wall components such as (lipo)teichoic acids, neutral and acidic polysaccharides, peptidoglycan and (surface) proteins. Another recent study reported that peptidoglycan of Lactobacillus paracase has anticancer activity by inducing caspase-3 activation in vitro, and that the combination of Bifidobacteria lipo-teichoic acid and 5-FU significantly inhibits tumor proliferation and induces apparent apoptosis [36, 37]. In this present study, it is assumed that the anti-tumor effect of that heat-killed L. casei MG4584 and L. reuteri MG5346 is due to lipoteichoic acid, pepti-doglycan, and polysaccharides etc., and further studies are needed to confirm this.

Reviewer 2 Report

This study aimed to investigate the therapeutic properties of heat-killed Lactobacillus strains against human colorectal cancer. The positive control is the Bifidobacterium bifidum MG731. The authors assessed several strains:  a) their cytotoxic effects in vitro on RKO cells, b) their apoptotic effects in vitro on RKO cells, c) their anti-tumor effects in RKO xenograft model. The methods are accurate and well described. They found important anti-tumor effects especially with parabiotic mixtures. These effects are mediated through increased apoptosis, as cleaved caspases 3 and cleaved PARP were higher in tumors of mice treated with parabiotics. The results are interesting and promising, even if the molecular mechanisms are not studied in depth. These findings are new and the manuscript is well written.

I have very minor comments.

Title: Only one xenograft model is used. We are expecting more when we read the title. Can you just remove the “s”? Same comment for the first sentence of the abstract.

Title: The most validated effect is for caspase 3. Why did you choose to highlight caspase-9 in the title?

Sorry but I don’t understand the last sentence of the abstract.

The statistical significance threshold is indicated by an * in all figures. Please correct in the caption of all the figures “different letters indicate significant differences”

I would suggest to define the parabiotic mix groups in the text at the beginning of the results, even if it is done in the figure captions.

Line 46. This suggests a clinical effect or a therapeutic effect in vivo but the reference is an in vitro study. Please be more specific.

Line 47-48. This is very important and there is no reference. Same comment for line 54.

Author Response

Reviewer #2

This study aimed to investigate the therapeutic properties of heat-killed Lactobacillus strains against human colorectal cancer. The positive control is the Bifidobacterium bifidum MG731. The authors assessed several strains:  a) their cytotoxic effects in vitro on RKO cells, b) their apoptotic effects in vitro on RKO cells, c) their anti-tumor effects in RKO xenograft model. The methods are accurate and well described. They found important anti-tumor effects especially with parabiotic mixtures. These effects are mediated through increased apoptosis, as cleaved caspases 3 and cleaved PARP were higher in tumors of mice treated with parabiotics. The results are interesting and promising, even if the molecular mechanisms are not studied in depth. These findings are new and the manuscript is well written.

I have very minor comments.

Title: Only one xenograft model is used. We are expecting more when we read the title. Can you just remove the “s”? Same comment for the first sentence of the abstract.

à We removed all erroneously included 's' from the paper, including the title and abstract.

Title: The most validated effect is for caspase 3. Why did you choose to highlight caspase-9 in the title?

  • We wanted to emphasize caspase-9 because caspase-3 and PARP are the executive caspases in apoptosis and caspase-9 is the initiator caspase that causes intrinsic apoptosis. Therefore, we revised the paper to emphasize this part.

Sorry but I don’t understand the last sentence of the abstract.

à We corrected the last sentence of the abstract.

The statistical significance threshold is indicated by an * in all figures. Please correct in the caption of all the figures “different letters indicate significant differences”

  • We corrected all incorrect sentences in figure legends.

I would suggest to define the parabiotic mix groups in the text at the beginning of the results, even if it is done in the figure captions.

  • We defined the parabiotics mixture group at the beginning of the results based on the reviewers' comments.

Line 193-198: To confirm the anti-tumor effect of heat-killed Bifidobacterium and Lactobacillus, we orally administered 1 × 109 cells / mouse of B. bifidum MG731, L. reuteri MG5346, L. casei MG4584, respectively in single strain groups of xenograft model. The 2Mix group was administered mixture of L. reuteri MG5346 (1 × 109 cells / mouse) and L. casei MG4584 (1 × 109 cells / mouse), and the 3Mix group was administered mixture of B. bifidum MG731 (1 × 109 cells / mouse), L. reuteri MG5346 (1 × 109 cells / mouse), and L. casei MG4584 (1 × 109 cells / mouse).

Line 46. This suggests a clinical effect or a therapeutic effect in vivo but the reference is an in vitro study. Please be more specific.

à The reference is clinical trials, we revised the paper to be more specific based on the comments of the reviewers.

Line 267-262: Previous studies have reported that the administration 2 × 109 CFU of live probiotic strains in CRC patients for 12 weeks improved bowel symptoms and quality of life [21]. Moreover, pre-surgical use of Lactobacillus acidophilus, Lactobacillus casei, Lactoba-cillus lactis, Bifidobacterium bifidum, Bifidobacterium longum, and Bifidobacterium infantis mixture promotes faster return of normal gut function and shorter duration of hospital stay when used in elective surgery in colorectal cancer patients [22].

Line 47-48. This is very important and there is no reference. Same comment for line 54.

  • We added references as you suggested.